# Effectiveness of an Acrobatic Gymnastics Programme for the Improvement of Social Skills and Self-Esteem in Adolescents

Xoana Reguera-López-de-la-Osa [1,*], Luis Arturo Gómez-Landero [2], Pureza Leal-del-Ojo [3] and Águeda Gutiérrez-Sánchez [1]

1    Education, Physical Activity and Health Research Group (Gies 10-DE3), Galicia Sur Health Research, Institute (IIS Galicia Sur), SERGAS-UVIGO, 36208 Vigo, Spain

2    Physical Performance & Sports Research Center, Universidad Pablo de Olavide, Ctra. de Utrera, km. 1, 41013 Seville, Spain

3    Department of Sports and Computer Sciences, San Isidoro University Center, Cartuja Island, 41092 Seville, Spain

*    Correspondence: xreguera@uvigo.es

**Abstract:** Background: The study of psychosocial aspects in adolescents is of increasing interest in the field of formal education. Therefore, the design and evaluation of an Acrobatic Gymnastics (AG) programme focused on the personal and social needs of adolescents in Physical Education (PE) is proposed. The objectives of this research are to establish the associations and relationships between self-esteem and social skills in order to determine the effectiveness of the programme. Methods: This is an evaluative research study in which a participatory action research method was used. Eighty-three secondary school students aged 14 and 15 participated in the study. The Ambez@r Group Social Skills Questionnaire and the Rosenberg Self-Esteem Scale were used for data collection. Results: Students with socially skilful behaviour have a high level of self-esteem. The designed programme produces statistically significant improvements in both constructs. Conclusions: The programme is effective in improving the social skills and self-esteem of adolescents, and there is a clear association between both constructs.

**Keywords:** acrosport; gymnastics; cooperative learning; physical education; secondary education; social and personal skills



## 1. Introduction

Adolescents find themselves in a society that is increasingly influenced by values such as individualism, success, competitiveness and aggressiveness, thereby leading them to behave capriciously, with little acceptance of rules, and manifesting violent episodes when relating to others [1]. On the other hand, the need to feel accepted in a group generates negative emotions that interfere with their personal well-being [2].

Educational centres propose the development of skills to increase democratic values, cooperation and tolerance in order to improve coexistence in the classroom, so that adolescents can integrate into the world of work and society as well as exercise their rights and obligations as citizens [3].

The relationship and behavioural problems that adolescents present in the classroom are a consequence of physical, affective, cognitive and social changes, making group cohesion and the teaching–learning process difficult. This leads adolescents to present disruptive behaviours [4–6].

Throughout the process of personality formation, several social skills must be acquired that enable good communication and guarantee success when it comes to relating to others. Social skills are acquired progressively and change over the course of a lifetime depending on the learners' stage of their educational process [7]. Their acquisition and development occur through a combination of personal maturity and the learning experiences offered in

educational centres [8]. These spaces prepare the individual for life, seeking to equip them with attitudes and values that allow them to adapt to the needs of the environment and generate a good climate of coexistence, being competent to participate in group activities, without showing social prejudice and valuing the differences of others [7,9–11].

Studies on prosocial behaviour have determined that it exerted an inhibitory effect on maladaptive social interaction styles, such as aggressiveness and social withdrawal, playing a fundamental role in the formation of positive interpersonal relationships and acceptance by peers, parents and teachers [12–14]. Prosocial behaviour has been shown to be closely related to study motivation and academic success [7,12,15]. There are studies that show a clear relationship between social and personal skills, concluding that an improvement in these skills has a positive impact on the development of adaptive, problem-solving, stress and frustration management skills [7,11,16,17]. Therefore, the development of personal and social skills as a key factor in promoting adolescents' social and academic competence should be considered.

Self-esteem is a personal skill that can be defined as an evaluation, state of mind or attitude that one has about oneself that is related to personal beliefs about one's abilities, social relationships and achievements [18–20]. Studies on self-esteem at school focus on the well-being of personal health. It has been shown that adolescents with low self-esteem are prone to underperform academically and display attitudes of hostility, reluctance and giving up [20]. In a sample of 302 students wherein self-esteem levels were measured in relation to academic performance, it was observed that high achievers scored significantly higher on self-esteem than low achievers. Their stimulation has therefore a positive impact on academic performance and conflict resolution [19].

This information is provided by the self-concept, which not only indicates how one sees oneself physically, academically or from a social perspective, but also from a personal perspective, and by the feeling of being of worth as a person, which is provided by self-esteem [21–23]. Personal skills are one of the most important variables within the motivational domain [24], which has a significant impact on the correct functioning of the cognitive domain. In this sense, Goudas, Dermitzaki and Bagiatis [25] argue that a teaching style that provides students with the opportunity to choose, participate and make decisions makes participation more enjoyable by influencing students' intrinsic motivations.

Different studies carried out in the field of martial arts and contact sports have high-lighted the figure of the teacher as a guide and educator, providing greater importance to the cultivation of human and social values than to the sport results due to the impact they have on the somatic, mental and social dimensions of health [26–28]. These and other similar studies have validated the connection between practising these sport disciplines and the development of personality, willingness, commitment to achieving one's goals, perception of the world and interpersonal relationships. All these studies pointed out the importance of shared work and student involvement in assessment processes for a better understanding of personal learning [11,16,27,29,30].

Thanks to the concern of social psychology to work along the lines of integration, respect and cooperation with others, new methodological proposals emerge in the area of Physical Education (PE) that facilitate personal relationships [13,31–34].

Research has shown the effectiveness of cooperative learning (CL) over more traditional instructional models in promoting students' academic achievement, improving classroom climate and working on conflict resolution in different curricular areas and educational stages. This is a PE-legitimate pedagogical model that favours inclusion as it is able to facilitate learning in the physical, cognitive, social and affective domains [10,15,31,34–36]. In this sense, the key to the success of this pedagogical model lies in the CL structures used as they favour students' autonomy, socialisation, trust, communication and the ability to encourage as well as solidarity and motivation [37–39]. This is achieved by having to complete a common task through dialogue, searching for information or using the necessary resources to achieve the final objective [40]. In addition, the positive effects of CL

compared to traditional forms of PE on self-esteem have been demonstrated as self-esteem is positively related to success, mental health and sport practice [41].

Formative assessment is given an important value in relation to methodological changes in education for a global transformation of the teaching–learning process [42]. The research results on the use of peer assessment have shown an increase in the level of motivation, perceived teacher confidence and competence as well as teacher self-efficacy [43]. A study on self-assessment as a strategy for collaborative learning in gymnastics concluded that the inclusion of students in assessments allows them to participate in the teaching–learning process and can contribute to the formation of more critical and autonomous individuals [44].

In order to achieve the optimal development of students' physical, cognitive, affective and social potentials, students must be part of the teaching–learning process so that they can self-regulate their progress, know the reason for the activities they carry out and understand the evaluation criteria that will verify learning. This is achieved through peer assessment processes, wherein students assess their peers, acquiring a role of observer and evaluator that broadens their vision of the teaching–learning process [43].

PE is a curricular area developed outside the students' regular classroom in which a variety of groupings are allowed and interaction is required, thus fostering various values such as solidarity, companionship, tolerance, empathy and respect. This leads to an increase in self-esteem, improved social relations, reciprocal learning among students and responsibility through motor actions [39,44–46].

Acrobatic Gymnastics (AG) is didactic content that allows working in groups of AC. This is an eminently cooperative sport that consists of producing a chain of collective and individual acrobatic forms within a choreography with musical support [32,47]. The pursuit of common goals, as in the case of AG, requires students to work in groups, to help and support each other, to be aware of the importance of each other in the successful creation of the figures and to listen and share ideas [45]. AG improves the levels of self-concept in those subjects with physical and/or emotional problems, as the students, by feeling involved in the activity they are doing and cooperating with their classmates, feel physically and emotionally enriched [46].

PE teachers perceive the development of gymnastic skills as something that could guarantee the integral development of students, favouring the acquisition of attitudes such as responsibility, effort, confidence and self-esteem [32].

Due to the evolutionary development of adolescents and the relationship and behavioural problems that arise in the classroom, the design and evaluation of an AG programme focused on their personal and social needs within the area of PE is proposed. The programme seeks to respond to these needs through activities that improve group cooperation, responsibility, communication, social interaction, creativity and motor competence. Hence, the aim of this research is to improve adolescents' social skills and self-esteem as well as to establish the associations and relationships between these two attributes in order to determine the effectiveness of the programme.

## 2. Materials and Methods

### 2.1. Design

This is an evaluative research study conducted from a critical perspective, using a participatory action research method with quantitative techniques. This type of research has been selected because it is a psychosocial study focused on enhancing the social and personal skills of adolescent students with the aim of improving a situation in the educational environment. Action research is a suitable process for research in the contexts of improving professional practice when carried out by the professionals involved [48].

### 2.2. Participants

The sample consisted of 83 secondary school students from the region of Galicia, Spain, of whom 45 (54.9%) were male and 37 (45.1%), female, aged between 14 and 15 years old. The participants' mean age was 14.60 with a standard deviation of 0.83. (Table 1).

**Table 1.** Distribution of the sample.

|  | **N** |  | **F** | **%** | **$\overline{X}$** | **SD** |
|---|---|---|---|---|---|---|
| Gender | 82 | Male | 45 | 54.9 |  |  |
|  |  | Female | 37 | 45.1 |  |  |
| Age | 82 | 14 years | 42 | 51.2 | 14.60 | 0.83 |
|  |  | 15 years | 40 | 48.8 |  |  |

There were no students in the sample who had special educational needs with curricular adaptations. They were heterogeneous groups with great differences in terms of physical fitness levels, motor skills and academic interests. They stand out for having difficulties relating to each other and for being very competitive. They are difficult to work with because they fear criticism and judgement. They also show difficulties in working in groups that are not made up of like-minded members. It is important to note that this was the first time they had participated in CL experiences.

The selection criterion for the inclusion in the sample was being enrolled in the second cycle of secondary education in the same school. On the other hand, the only exclusion criterion was the families' failure to sign the informed consent form. There were no exclusions, as all the families signed the consent forms.

### 2.3. Instruments

In order to determine social skills, the Social Skills Questionnaire of the Ambez@r Group [49] was used. This instrument measures the percentage of socially skilful behaviour perceived by the pupils. It consists of a total of 20 situations in which a subject may find themselves. The behaviours shown determine the greater or lesser degree of adaptation with respect to the groups with which the subject relates, which will make them feel more or less accepted, relaxed or show greater or lesser happiness. The respondent must evaluate according to the number of times they find themselves in the different situations described. The frequency and evaluation of the instrument correspond to the value 2, almost always; 1, sometimes; and 0, almost never.

For the analysis of the self-esteem construct, the Rosenberg Self-Esteem Scale [23] was used. The Spanish version consists of 10 items, focusing on feelings of self-respect and self-acceptance. They are rated on a four-point Likert scale (1 = Strongly Agree/4 = Strongly Disagree), providing a total score ranging from 10 to 40 points. Half of the items are stated positively and half negatively to control for the effect of acquiescence. For items 1 to 5, responses are rated 4 to 1 and for items 6 to 10, responses are rated 1 to 4. From 30 to 40 points, self-esteem is considered to be normal; a score from 26 to 29 points indicates medium self-esteem, which should be improved; and a score less than 25 points means that self-esteem is low.

Table 2 shows the summary statistics of the scales, indicating the total of each construct and showing their reliability for the pre-test and post-test measurements of each variable.

**Table 2.** Summary statistics and reliability of the social skills and self-esteem scales.

| **Variables** | **$\overline{X}$** | | **SD** | | **Cronbach's Alpha** | |
|---|---|---|---|---|---|---|
|  | **Pre** | **Post** | **Pre** | **Post** | **Pre** | **Post** |
| Social Skills | 1.43 | 1.50 | 0.38 | 0.32 | 0.56 | 0.65 |
| Self-esteem | 3.03 | 3.17 | 0.70 | 0.54 | 0.88 | 0.89 |

Cronbach's alpha coefficient was calculated to analyse the reliability of both scales. A result equal to or higher than 0.7 is considered reasonable or satisfactory.

The results obtained for the social skills questionnaire in the pre-test measure were = 0.56 and in the post-test, =0.65, showing a higher value in the final assessment, with values closer to 0.7. However, the reliability of the scale is considered low. Despite these results, it was decided to use this questionnaire because it was the one employed by the school guidance personnel and the students were familiar with it.

As for the results obtained for self-esteem, it was observed that, in the pre-test measure, a value of 0.88 was obtained and in the post-test, a value of 0.89. The original scale had an alpha coefficient of 0.96. In addition, it was evaluated in different countries and languages and with different populations, finding in all of them adequate psychometric properties of the instrument, with values, in general, between the ideal coefficients 0.70 and 0.90 [18,23,50].

*2.4. Procedure*

Action research makes it possible to endow educational research with rigour [51]. To this end, a process of cycles designed on the basis of introspective spiral proposals was followed.

1st cycle

- Preliminary phase: a diagnostic analysis, a review and a status of the issue in different documentary sources were carried out in order to design the pilot study.
- Programme design phase: the programme was designed, implemented and evaluated. After analysing the results and identifying its weaknesses, it was modified to be carried out in the next cycle.

2nd cycle

- Programme implementation phase: The same as during the implementation of the pilot study, permission was sought from the management team and informed consent was obtained from the families in order to send them the questionnaires before and after the implementation of the programme. The methods used complied with the deontological standards recognised by the Declaration of Helsinki (with the latest revision approved in Brazil, October 2013), in accordance with the recommendations of Good Clinical Practice of the EEC (CPMP/ICH/135/95 of July 2002) and the current Spanish legal regulations governing research.
- Final evaluation phase: the analysis of the results and the drafting of the report were carried out, and the limitations of the study were pointed out in order to be taken into account in the next cycle. We are currently in the phase of dissemination and impact assessment.

This programme has been designed to be implemented in the educational environment, and so it responds to the training requirements stipulated by the Competence Definition and Selection Project (DeSeCo) [52] in terms of the contribution of key competencies involving the mobilisation of practical and cognitive skills, creative abilities and other psychosocial resources, such as attitudes, motivation and values.

The legislation on which it is based is the Organic Law, 8/2013, of 9 December, for the Improvement of the Quality of Education [53] and Decree 86/2015, of 25 June, which establishes the Compulsory Secondary Education Curriculum in the Autonomous Community of Galicia (Spain) [54].

It consists of an integrated didactic unit taught during the third term of class. It is made up of eighteen sessions (two 50-min sessions per week) in which the content blocks of Physical Condition and Health as physical conditioning, AG as an individual sport in Games and Sports, and Corporal Expression through choreographic and musical interpretation are developed.

In the Physical Condition and Health block, the contents relating to the ability to carry out a task, physical health and physical exercise as well as sports habits are grouped

together, which affect motor development and people's ability to improve their quality of life.

In the Games and Sports block, the skills pursued by this subject are promoted through the development of specific skills through the development of attitudes aimed at solidarity, cooperation and non-discrimination.

Finally, the Body Expression block incorporates content related to knowledge and awareness of the body, its possibilities of movement in time and space as well as the use of different body techniques as a way of learning to express and communicate emotions, feelings and ideas and as a means of developing the skills of relaxation, concentration, breathing, disinhibition and awareness.

As a pedagogical model, CL has been chosen as the search for individual and group success allows adolescents to learn to relate to each other by improving their communication, thus assuming their roles within groups in order to contribute the best of themselves and to learn from everyone while everyone wins and has fun [36,38].

The following CL structures were used: reciprocal teaching for learning the gymnastic elements [55]; think-share-act for the pyramid initiation activities [56]; Learning Team for doing the work [56]; and jigsaw for learning to design choreographies as a group [57]. The working groups were composed by adolescents following the requirements established by a teacher [58].

As a learning strategy, a tutored learning project was used, and as a didactic resource, they were given a dossier made up of the necessary material for the development of the didactic unit. Formative and shared evaluation processes followed.

The following structured and systematic instruments were used: a teacher's anecdotal record; a pupils' class diary and the dossier with the observation sheets, the technical sheet for the observation of the choreography, the descriptive scales and a script for the final reflection. As an unstructured instrument, the last five minutes were dedicated to reflecting on what had been worked on and another five minutes, for them to record the relevant information in their diaries.

In order to meet the needs as well as develop the potentials and interests of the students, an atmosphere of affection and trust was created, which facilitated the teaching–learning process and the final reflections carried out in the final assembly.

### 2.5. Data Analysis

For treatment of the data, normality was assumed (N = 82), and so the tests applied were parametric, with a confidence interval of 95%.

The Student's *t*-test and analysis of variance (ANOVA) were used to compare the mean scores as along with Pearson's correlation to observe the relationships between the scores of the different instruments used. The effect size was analysed using Cohen's d with the following values: d < 0.2 (null), d = 0.2–0.49 (small), d = 0.5–0.80 (moderate) and d > 0.8 (large). The statistical analysis of the research was carried out with the IBM SPSS Statics 25 statistical package.

## 3. Results

### 3.1. Association between Self-Esteem and Social Skills

In order to test the relationship between the variables self-esteem and social skills, an ANOVA was calculated (Table 3). After observing that there was significant evidence ($p < 0.05$) between both variables, Bonferroni was calculated to determine the degree to which one influenced the other.

The analysis of variance showed significant differences between self-esteem and social skills, both in the pre-test ($p = 0.024$) and post-test ($p = 0.031$). The Bonferroni analysis showed that social skills are associated with a high level of self-esteem (pre-test: $p = 0.022$ and post-test: $p = 0.049$). This indicates that students with socially skilful behaviour have a high level of self-esteem.

**Table 3.** ANOVA and Bonferroni for self-esteem grouped with the dependent variables.

| Self-Esteem | | N | $\overline{X}$ | F | *p* | Bonferroni | |
|---|---|---|---|---|---|---|---|
| Pre-test Social Skills | Low | 18 | 65.83 | | | | |
| | Medium | 15 | 72.50 | 3.906 | 0.024 | Low/High | 0.022 |
| | High | 49 | 73.27 | | | | |
| | Total | 82 | 71.49 | | | | |
| Post-test Social Skills | Low | 18 | 71.11 | | | | |
| | Medium | 15 | 72.00 | 3.626 | 0.031 | | |
| | High | 49 | 77.60 | | | Low/High | 0.049 |
| | Total | 82 | 75.15 | | | | |

*3.2. Correlation Analysis*

To determine the correlation between the constructs, Pearson's correlation coefficient was calculated (Table 4). The correlational analysis showed a high correlation and positive effective selection between the pre-test and post-test of social skills (0.76) and self-esteem (0.85). This indicates that the constructs are highly and positively correlated with each other.

**Table 4.** Pearson's correlation between the analysed constructs.

| | | Social Skills | | Self-Esteem | |
|---|---|---|---|---|---|
| | | **Pre** | **Post** | **Pre** | **Post** |
| Post Social skills | $\Upsilon$ | 0.759 ** | | | |
| Pre Self-esteem | $\Upsilon$ | 0.398 ** | 0.358 ** | | |
| Post Self-esteem | $\Upsilon$ | 0.406 ** | 0.365 ** | 0.850 ** | |

Correlation significant at ** $p < 0.01$ levels.

In the analysis conducted for the correlation between social skills and self-esteem, it was observed that there was a moderate correlation and an effective and positive selection between the pre-test of social skills and self-esteem (0.40), a low correlation and a positive selection between the post-test of social skills and the pre-test of self-esteem (0.36), a moderate correlation and a positive selection between the post-test of social skills and the pre-test of self-esteem (0.36), a moderate correlation and positive selection between the social skills pre-test and the self-esteem post-test (0.40) as well as a low correlation and positive selection between the social skills and self-esteem post-test (0.37).

Therefore, the correlational analysis of both constructs showed an overall pattern of high correlation with each other and moderate-low correlation between the constructs of social skills and self-esteem.

*3.3. Descriptive Analysis*

In order to test the effectiveness of the programme, a descriptive analysis of the variables self-esteem and social skills was carried out according to the time of administering the instruments. A Student's *t*-test was calculated for related samples (N = 82) with a confidence interval of 95% (Table 5). It was observed that skewness and kurtosis were negative, indicating that the values were more clustered at levels above the arithmetic mean and, therefore, negative kurtosis indicates a lower concentration of data around the mean.

It can be seen that there was an improvement in the difference in means obtained between the pre-test and post-test for social skills (post-pre: 3.66) and self-esteem (post-pre: 1.36) after the intervention in the classroom.

Significant differences ($p < 0.05$) are evident for the constructs of social skills ($p < 0.001$) and self-esteem ($p < 0.001$). This indicates that the designed programme produces statistically significant improvements in both. The effect size was analysed using Cohen's d with small values for both differences (d = 0.35 for social skills and d = 0.25 for self-esteem).

**Table 5.** Descriptive analysis of the variables and Student's *t*-test for related samples (N = 82).

| Variables | $\overline{X}$ | SD | $\overline{X}$pos-Pre | Asim. | Curt. | Min. | Max. | t Student | | |
|---|---|---|---|---|---|---|---|---|---|---|
| | | | | | | | | t | p | Cohen's d |
| Pre Social skills | 71.49 | 10.11 | | −0.06 | 0.12 | 45 | 98 | | | |
| Post Social skills | 75.15 | 10.40 | 3.66 | −0.47 | −0.15 | 50 | 95 | −4.648 | 0.000 | 0.356 |
| Pre Self-esteem | 30.34 | 5.38 | | −0.32 | −0.52 | 17 | 40 | | | |
| Post Self-esteem | 31.7 | 5.36 | 1.36 | −0.51 | 0.26 | 16 | 40 | −4.204 | 0.000 | 0.253 |

## 4. Discussion

The purpose of this research was to evaluate the effects of an AG programme in a PE classroom focused on the improvement of adolescents' social skills and self-esteem. In relation to the association between self-esteem and social skills, a high correlation between the pre-test and post-test means of each construct is demonstrated. This indicates that when an adolescent has an acquired ability and this is reinforced, it increases significantly. In this sense, similar research verifies that high self-esteem correlates with prosocial behaviours in adolescents [7,17,59].

The analysis of the results shows a significant improvement in social skills and self-esteem after the implementation of the programme. Based on the literature analysed, it can be claimed that these results are justified by the teaching methodology used and the formative and shared assessment processes followed. In fact, CL favours social interaction and student inclusion [60]. CL has been effective in favouring this situation, making it an excellent methodological resource for fostering a positive classroom climate, favouring coeducation, improving socialisation, the capacity for dialogue and adolescent cognitive and motor skills [31,34,45,60–62].

AG is considered to be an excellent cooperative and integrative proposal for motor and attitudinal development [37,63]. Moreover, Physical Education teachers identify this modality as the most suitable for developing personal and social values in students and say that it can favour their teaching–learning process due to its high collaborative component, thus minimising the personal differences between classmates [32]. In the study carried out by Piepiora, Piepiora and Bagińska [16] on the relationship between sport practice and personality development, they concluded that the more sport, the greater the openness to experience, kindness and awareness, considering the importance of establishing educational proposals based on the formation of personality from PE classes.

In a study carried out in the field of martial arts, an analysis of the socio-cultural factors that determined the opposition between the role of a martial arts teacher and that of a sport coach was carried out. It concluded that inevitably a teacher must be both a teacher and an educator [26]. In another study, with similar characteristics based on combat sports, similar conclusions were drawn, with most coaches preferring to focus on educational values rather than on sport achievements. The cultivation of these values may play an increasingly important role in expanding the health dimensions (somatic, mental and social) of young people [28]. We should keep in mind that the figure of the teacher, in our case, requires a guiding role during the teaching–learning process that goes beyond the mere transmitter of concepts. Thus, a PE teacher should organise and plan their classes in such a way that seeks to meet the social and personal needs of adolescents so that the teaching–learning process can be carried out in the most effective way possible, thus improving the classroom climate.

In this sense, and in relation to another study in the field of martial arts and combat sports [27], the coaches or teachers of these sports should bear in mind that the purpose of their work should focus on creating an environment of mental and social self-cultivation through a PE curriculum, an opportunity provided to us through the content of AG in this same educational area.

In the study carried out by Ávalos-Romero, Grande and Vega [44] on self-assessment as strategies for collaborative learning of gymnastics, they concluded that practising AG brings many affective–social benefits. In this aspect, the inclusion of students in the

evaluation processes was key, as it allows them to participate in their teaching–learning process, contributing to the formation of more critical and autonomous people. Research in the field of PE has shown that peer assessment is a shared assessment mechanism that encourages student participation and promotes learning. In these studies, positive results were obtained in relation to the increase in motivation to learn and the level of confidence in secondary school students [42,43]. Students' involvement in assessment processes enables them to better understand the learning process and to identify their progress in relation to their educational goals. Thus, teachers and students can better determine their progress in relation to their personal educational goals [27].

Gymnastic skills promote personal and social values and abilities in students, such as self-esteem, responsibility, confidence, cooperation, teamwork and self-management of fears and insecurities [32,64]. Previous research has found that AG is an appropriate modality for promoting values among adolescents. Its collaborative, group and inclusive nature allow for minimising the personal differences among students [32,45,65].

The feeling of social acceptance and personal worth by being able to perform gymnastic elements, pyramids and compose a group choreography are perceived by adolescents as a sense of competence and worth that improves their self-esteem [66]. Other research has found that the content of AG favours the development of social skills and self-esteem [32,45,67,68].

In this research, CL allowed the participation and involvement of each of the group members in achieving the final objective, creating an AG choreography, to improve the students' levels of self-esteem [45,69,70].

Several researchers conclude their studies by reiterating the importance of designing programmes aimed at developing the basic components of social cohesion in the most important contexts of socialisation: the family and the school [7,14,71]. To this end, social bonds, trust, collective knowledge, sense of belonging, values and feelings must be emphasised. All these basic components have been worked on with the designed programme, obtaining a positive result in the improvement of social skills and self-esteem.

Regarding the limitations of this study, it should be noted that the sample is relatively small, as it was carried out in a single school. It would be advisable to implement this programme in other schools with different characteristics, expanding not only the population under study, but also its diversity. Another limitation is that there was no analysis of the influence of gender on social skills and self-esteem. A study following this line of research is suggested, as it could provide interesting data on how to work with these constructs in the educational context.

## 5. Conclusions

In conclusion, it was found that the AG programme is effective in improving the social skills and self-esteem of adolescents, and there is a clear association between the two constructs.

The CL is a didactic methodology that favours the creation of a positive classroom climate in which adolescents feel integrated within a group. This makes them improve their prosocial behaviours and feel better about themselves.

The content of the AG favours cooperative work and improves individual and group confidence, not only improving adolescents' social skills, but also giving them emotional stability and improving their self-esteem.

**Author Contributions:** Conceptualization, X.R.-L.-d.-l.-O., Á.G.-S., L.A.G.-L. and P.L.-d.-O.; methodology, X.R.-L.-d.-l.-O., Á.G.-S., L.A.G.-L. and P.L.-d.-O.; software, X.R.-L.-d.-l.-O. and Á.G.-S.; validation, X.R.-L.-d.-l.-O. and Á.G.-S.; formal analysis, X.R.-L.-d.-l.-O. and Á.G.-S.; investigation, X.R.-L.-d.-l.-O. and Á.G.-S.; resources, X.R.-L.-d.-l.-O. and Á.G.-S.; data curation, X.R.-L.-d.-l.-O. and Á.G.-S.; writing—original draft preparation, X.R.-L.-d.-l.-O. and Á.G.-S.; writing—review and editing, X.R.-L.-d.-l.-O., Á.G.-S., L.A.G.-L. and P.L.-d.-O.; visualization, X.R.-L.-d.-l.-O. and Á.G.-S.; supervision, X.R.-L.-d.-l.-O., Á.G.-S., L.A.G.-L. and P.L.-d.-O.; project administration, X.R.-L.-d.-l.-O. All authors have read and agreed to the published version of the manuscript.

**Funding:** This research received no external funding.

**Institutional Review Board Statement:** The research plan was approved by the Academic Committee of the Doctoral Program in Sports Education and Health at the University of Vigo (CAPDEDS-27/09/2017). The Academic Committee and the researchers confirm that research using human participants has been conducted ethically according to the principles of the Declaration of Helsinki (revision of Fortaleza, Brazil, 2013) and in accordance with the recommendations of Good Clinical Practice of the EEC (document 111/3976/88 of July 1990) and the current Spanish legal regulations on research, as well as the AERA standards.

**Informed Consent Statement:** Informed consent was obtained from all the subjects involved in the study.

**Data Availability Statement:** Not applicable.

**Acknowledgments:** We would like to thank the school's management, teachers, students and families for their collaboration in the development of this research study.

**Conflicts of Interest:** The authors declare no conflict of interest.

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
