# Peer review of "Effectiveness of an Acrobatic Gymnastics Programme for the Improvement of Social Skills and Self-Esteem in Adolescents"

_sustainability, doi:10.3390/su15075910_

Round 1

Reviewer 1 Report

I have included my comments and suggestions in the attached file.

I cordially greet you, Dear Authors.

Author Response

A revision was suggested to improve the theoretical basis of the introductory part. We thought this was a good observation and we took it into account. In his revision he suggested the inclusion of some references. These were read and revised with the aim of being included in both the introduction and the discussion.
In some cases we find it difficult to find a direct relationship with the object of our study, since Acrobatic Gymnastics is an eminently cooperative sport where students act simultaneously with other students to reach a common goal. Unlike martial arts, where the performance is solo and in front of an opponent. On the other hand, both in the legislation that regulates secondary education in our country, as well as in the programming of the educational centre, combat sports are not included. However, we hope that you will be pleased with their inclusion.
The aim of the last paragraph of the introduction has been modified for a better clarity of the main question of this research.

Reviewer 2 Report

The article is well constructed, clearly presented and with arguments, both of the concepts used and of the theoretical approach, but also of the research design.

The authors make detailed statements on the state of knowledge, referring to sources on the topic, including the Spanish sample, the cited sources are consistent. One feels the need for a clearer argumentation of what this article brings, but also of originality.

Also, some clarifications related to the representativeness of the sample, as well as the characteristics of the subjects (for example, can gender influence the level of self-esteem and social skills?). Maybe also here with the subjects, some results of the statistical techniques should be detailed to highlight the effects of these independent variables on self-esteem and social skills. If other results are introduced, they should be analyzed within the Discussions.

Author Response

Dear reviewer, thank you very much for your positive feedback and input. The justification section of the article was improved by arguing what this article contributes. The information regarding the sample and the characteristics of the subjects was expanded. We found the contribution on the influence of gender on self-esteem and social skills very interesting. It has not been included in this study due to the orientation of the article. We did not take into account the gender difference as an objective of the study, since it seemed more interesting to focus only on the AG programme and the two variables analysed, but we will take it into account for future research and include it as a limitation of the study.

Reviewer 3 Report

Dear authors, 

Below I provide some suggestion in order to review your manuscript:

-       I would recommend including more information regarding the study participants. In particular, inclusion-exclusion criteria and details such as whether they belong to the same school and/or class and if they have previously worked together. Similarly, the sample selection methodology should be detailed.

-       Although it is intuitive, the section detailing the instruments used in the study should have its own heading and title.

-       In the instruments section, it would be advisable to attach the reference to the Social Skills Questionnaire of the Ambez@r Group for the reader's reference.

-       Check in the table related to the instrument (Cronbach not Conbrach).

-       Regarding the instruments used, the use of the Social Skills Questionnaire should be justified in more detail because of the values achieved.

-       In the presentation of results, it would be interesting to add effect size values in addition to significance. In addition, in order to be able to enhance the analysis of the data, it would be interesting to know if there were any differences between the participants according to gender.

Author Response

We thank you for the suggestions made in your review. The following is a response to each of them:
- The information on study participants was expanded.
- We apologise and thank you very much for your comment. Section 2.3. was indeed missing, we must have deleted it when we transferred it to the template without realising it.
- The reference to the Social Skills Questionnaire was included and its choice and source was explained.
- Thanks again for identifying another typo (Cronbach not Conbrach).
- We added the effect size values.
- We find the contribution you made on gender differences among participants very interesting. These data were not included in the study due to the orientation of the article. We did not take into account the gender difference as an objective of the study, as we found it more interesting to focus only on the GA programme and the two variables analysed, but we will take it into account for future research and include it as a limitation of the study.

Reviewer 4 Report

This manuscript presents an interesting and novel approach towards the personal and social needs of adolescents through Acrobatic Gymnastics. The study Results and associated interpretation of these through the Discussion are satisfactory, there are many elements that need to be considered prior to these sections for the manuscript to be improved and at publication stage.

Abstract

- There is no mention of a control group. The statement is made that the programme was effective in improving social skills and self-esteem in adolescents, but in comparison to who?

Introduction

The Introduction is unclear, disjointed, and difficult to follow in many parts. The use of short one or two sentence paragraphs that are descriptive and lack connection between one another makes it difficult to establish a direction and underlying argument from the language.  I believe that considering the use of sub-headings throughout this section would be helpful for reader comprehension. There are many references used throughout, but these reflect statements made by the authors of the current manuscript, and little to no interpretation of what these statements mean - they are descriptive primarily. It would have been beneficial to include and incorporate examples to clarify and represent these statements, and to illustrate what is meant. For instance, "There are studies that show a clear relationship between social and personal skills, concluding that an improvement in these skills has a positive impact on the development of adaptive, problem-solving, stress, and frustration management skills". What social and personal skills are you referring to here, and what situations/circumstances/environments require adaptive, problem-solving, stress, and frustration management skills? Statements such as this and others require illustration and explanation for the audience.

- What is zone of proximal development (page 1, line 45)? In relation to what?

- How are thoughts and feelings of self-appreciation forged throughout life (page 2, line 63)?

- How does stimulation of self-esteem positively impact academic performance and conflict resolution? This needs to be evidenced (page 2, line 67).

- An example of cooperative learning would be useful (page 2, line 82).

-What is meant by "Due to the stage of change in which adolescents find themselves" (page 3, line 122)? What stage of change? All adolescents? This is a big statement.

Materials and Methods

 - Should say "This is an evaluative research study from a critical perspective". There is no justification as to why this methodology has been employed.

- There is no evidence of reliability and validity for the Social Skills Questionnaire or Rosenberg Self-Esteem Scale.

- What is a school guidance team (page 4, line 142)?

- Does Table 2 represent reliability from the sample in this study? What are M and DS in this Table? Are these Results?

- Statement about Action Research requires a reference (page 4, line 175).

The section on page 4 and page 5 outlining study procedure contains a lot of information; I believe this would be communicated more effectively with small sub-headings and larger paragraphs rather than scattered small paragraphs. I recommend reviewing this section to ensure that the language and explanations are clear, fluent, and sufficient detail of steps is provided. Throughout this section there is no detail provided regarding timelines for the study.

- What is the DeSeCo Project (page 5, line 194)?

Results

- What classifies LOW, MEDIUM, and HIGH representation.

- What is meant by "positive effective selection" (page 6, line 264)?

Overall, the Discussion section is sound, but I believe there is opportunity throughout to provide greater interpretation of the study findings and what they mean. There is strong communication of similar/comparable studies but highlighting study findings and the value these add to the field could be expanded.

Author Response

Dear reviewer, we thank you for your contributions, which we find very interesting. In the following, we provide answers to them:
- The effectiveness of the programme was done with the same group, comparing the results obtained before and after the intervention (pretest-posttest). There is no control group for several reasons. The first of these is that the study was carried out in a single school with a single line. Therefore, we did not have the option of using one class as the experimental group and another as the control group. Furthermore, the design of the programme seeks to respond to the requirements of the educational legislation that governs our educational system, which led us to implement a didactic methodology and a specific competence development that we consider positive for all students. And finally, because, although we use a control group, each class has a specific group of students with specific personal characteristics, so we understand that the dynamics of the class depends on the social relations of each group-class and the characteristics and personal needs of each student. It is very difficult to find two classrooms with the same number of students displaying disruptive behaviour, with similar classroom climates, peer relationships and personal characteristics.

Introduction

The Introduction is unclear, disjointed, and difficult to follow in many parts. The use of short one or two sentence paragraphs that are descriptive and lack connection between one another makes it difficult to establish a direction and underlying argument from the language.  I believe that considering the use of sub-headings throughout this section would be helpful for reader comprehension. There are many references used throughout, but these reflect statements made by the authors of the current manuscript, and little to no interpretation of what these statements mean - they are descriptive primarily. It would have been beneficial to include and incorporate examples to clarify and represent these statements, and to illustrate what is meant. For instance, "There are studies that show a clear relationship between social and personal skills, concluding that an improvement in these skills has a positive impact on the development of adaptive, problem-solving, stress, and frustration management skills". What social and personal skills are you referring to here, and what situations/circumstances/environments require adaptive, problem-solving, stress, and frustration management skills? Statements such as this and others require illustration and explanation for the audience.

- We have followed the rules of the journal and used the template with the established format. The journal does not offer the possibility of subtitles in the Introduction.
- We have joined some short paragraphs of the Introduction together for a better connection between them.
- Some interpretations of these statements have been included in the references.

What is zone of proximal development (page 1, line 45)? In relation to what? This has been specified in the document (p. 1 lines 44 and 45). The zone of proximal development is where learners are during the learning process. It is directly related to the established teaching-learning methods.

How are thoughts and feelings of self-appreciation forged throughout life (page 2, line 63)? As this sentence does not provide relevant information, it has been deleted and restructured with the following paragraph on self-esteem.

How does stimulation of self-esteem positively impact academic performance and conflict resolution? This needs to be evidenced (page 2, line 67). Clarified in that paragraph.

An example of cooperative learning would be useful (page 2, line 82). Done (p. 2, line 96-100).

What is meant by "Due to the stage of change in which adolescents find themselves" (page 3, line 122)? What stage of change? All adolescents? This is a big statement. The paragraph has been restructured to make it easier to understand (p. 3, line 135).

Materials and Methods

Should say "This is an evaluative research study from a critical perspective". There is no justification as to why this methodology has been employed. Done. (p 3, lines 145-150).

There is no evidence of reliability and validity for the Social Skills Questionnaire or Rosenberg Self-Esteem Scale. It is calculated. Table 2. It reflects the summary statistics and reliability of both scales. In addition, reliability studies of the Rosenberg self-esteem scale are added with their corresponding references (page 4, lines 186-203).

What is a school guidance team (page 4, line 142)? Such a team or department can be called by many names: school guidance personnel, school guidance department, School Guidance and Counseling..... (p. 5, line 197). In the paper we change the term to "school guidance personnel". Terminology used by the following article with 102 citations in scopus:
Muehlenkamp, J. J., Walsh, B. W., & McDade, M. (2010). Preventing non-suicidal self-injury in adolescents: The signs of self-injury program. Journal of youth and adolescence, 39, 306-314.

Does Table 2 represent reliability from the sample in this study? What are M and DS in this Table? Are these Results? Yes, shows the reliability results of this study. Refers to Mean (X) and Standard Deviation (SD). These acronyms have been modified.

Statement about Action Research requires a reference (page 4, line 175). Done (p.5, line 205).

The section on page 4 and page 5 outlining study procedure contains a lot of information; I believe this would be communicated more effectively with small sub-headings and larger paragraphs rather than scattered small paragraphs. I recommend reviewing this section to ensure that the language and explanations are clear, fluent, and sufficient detail of steps is provided. Throughout this section there is no detail provided regarding timelines for the study. Modifications were made in this section.

What is the DeSeCo Project (page 5, line 194)? Included in the manuscript. Refers to the Competence Definition and Selection Project. (p.5, line 229).

Results

What classifies LOW, MEDIUM, and HIGH representation. Explained in the manuscript (p.4, lines 178-188). This is the classification of self-esteem according to the ranges established by the Rosenberg Self-Esteem Scale. The word "self-esteem" was included in the table for ease of understanding.

And finally, following the recommendations of the other reviewers, several modifications were made in the Discussion section.

Reviewer 5 Report

First of all, I would like to thank you for the opportunity to review your work.
The topic you have taken up is still up-to-date in the science of physical culture.
However, your work requires corrections so that it can be processed further in the journal.
1. Remove the section numbers from the abstract.
2. Replace keywords with other than in the article title.
3. Remove chapter and subsection numbers.
4. Move the subsection "2.1. Design" as a sentence ending the chapter "Introduction".
5. Extend the description of the study population.
6. Create a subsection "Method".
7. Expand the discussion. Include the following articles:
in relation to the benefits of sport - Witkowski K et al. Social status of karate and personal benefits declared by adults practicing karate. Arch Budo Sci Martial Art Extreme Sport 2017; 13: 179-184; regarding the influence of the coach - Witkowski K et al. The role of a combat sport coach in the education of youth – a reference to the traditional standards and perception of understanding the role of sport in life of an individual and society. Arch Budo Sci Martial Art Extreme Sport 2016; 12: 123-130; in relation to sports experience - Piepiora P et al. Personality and Sport Experience of 20–29-Year-Old Polish Male Professional Athletes. Front. Psychol. 2022; 13: 854804. doi: 10.3389/fpsyg.2022.854804

Author Response

Dear reviewer, thank you for your positive feedback and input. We have responded to them:

- Section numbers were removed from the abstract.
- We replaced the keywords with different ones. We kept social skills, but not in isolation, but as social and personal skills.
- It is not possible to remove chapter and subsection numbers, as these are journal rules. We cannot remove them, and furthermore, other reviewers asked us to include some more subsections.
- Sorry, the subsection "2.1. Design" could not be included as the final sentence of the chapter "Introduction" as it is a section requested by the journal rules.
- The description of the study population was expanded.
- We are sorry, but each of the sections, subsections and sub-sections respect the journal guidelines. The template provided by Sustainability was followed.
- The references indicated in the introduction and discussion were inserted.

Round 2

Reviewer 1 Report

Dear Authors,

The manuscript has been corrected and supplemented. I believe that it can be accepted for publication in its current form.

Author Response

Thank you very much for your time and dedication in improving this article. The English has been proofread by a native speaker.

Reviewer 3 Report

Dear authors, 

Thank you for the improvements made to the scientific paper. 

Best regards.

Author Response

(The authors gave the same response as above.)

Reviewer 4 Report

Thank you for addressing my comments in a timely and professional manner. Your review and modifications have facilitated the manuscript now being easier to read and follow, logical in structure, and has provided both clarification and justification for several points.

I have only one comment which refers to the opening sentence of a paragraph on page 3, line 122:  good didactic material at allows working in CL groups and that contributes positively to aspects of human growth and development is AG.

To me this sentence is unclear and does not make sense, so I suggest considering this before the next stage of manuscript progression.

Author Response

Thank you very much for your kind words and the input on the sentence: good didactic material at allows working in CL groups and that contributes positively to aspects of human growth and development is AG (p. 3 line 122). It was modified by: Acrobatic Gymnastics (AG) is a didactic content that allows working in CL groups.

Reviewer 5 Report

The article as it stands meets the criteria of the journal and I recommend it for publication.

Author Response

(The authors gave the same response as above.)
